**Cite this article:** van Beest FM, Beumer LT, Chimienti M, Desforges J-P, Huffeldt NP, Pedersen SH, Schmidt NM. 2020 Environmental conditions alter behavioural organization and rhythmicity of a large Arctic ruminant across the annual cycle. R. Soc. Open Sci. **7**: 201614.

behaviour/ecology

Arctic tundra, behavioural rhythms, circadian organization, periodicity, polar vertebrates, ruminating herbivores

**Author for correspondence:**
Floris M. van Beest
e-mail: flbe@bios.au.dk

5179240.

# Environmental conditions alter behavioural organization and rhythmicity of a large Arctic ruminant across the annual cycle

Floris M. van Beest[1,2], Larissa T. Beumer[1,2],
Marianna Chimienti[1], Jean-Pierre Desforges[1,3],
Nicholas Per Huffeldt[1,4], Stine Højlund Pedersen[5,6]
and Niels Martin Schmidt[1,2]

[1]Department of Bioscience, Aarhus University, Frederiksborgvej 399, 4000 Roskilde, Denmark
[2]Arctic Research Centre, Aarhus University, Ny Munkegade 116, 8000 Aarhus C, Denmark
[3]Natural Resource Sciences, McGill University, Ste Anne de Bellevue, Quebec Canada, H9X 3V9
[4]Greenland Institute of Natural Resources, 3900 Nuuk, Greenland
[5]Department of Biological Sciences, University of Alaska Anchorage, Anchorage, AK, USA
[6]Cooperative Institute for Research in the Atmosphere, Colorado State University, Fort Collins, CO, USA

FMvB, 0000-0002-5701-4927; LTB, 0000-0002-5255-1889;
MC, 0000-0002-8236-9332; J-PD, 0000-0002-6816-7697;
NPH, 0000-0002-0154-2536; SHP, 0000-0003-0078-8873;
NMS, 0000-0002-4166-6218

The existence and persistence of rhythmicity in animal activity during phases of environmental change is of interest in ecology, evolution and chronobiology. A wide diversity of biological rhythms in response to exogenous conditions and internal stimuli have been uncovered, especially for polar vertebrates. However, empirical data supporting circadian organization in behaviour of large ruminating herbivores remains inconclusive. Using year-round tracking data of the largest Arctic ruminant, the muskox (*Ovibos moschatus*), we modelled rhythmicity as a function of behaviour and environmental conditions. Behavioural states were classified based on patterns in hourly movements, and incorporated within a periodicity analyses framework. Although circadian rhythmicity in muskox behaviour was detected throughout the year, ultradian rhythmicity was most prevalent, especially when muskoxen were foraging and resting in mid-winter (continuous darkness). However, when combining circadian and ultradian rhythmicity together, the probability of

behavioural rhythmicity declined with increasing photoperiod until largely disrupted in mid-summer (continuous light). Individuals that remained behaviourally rhythmic during mid-summer foraged in areas with lower plant productivity (NDVI) than individuals with arrhythmic behaviour. Based on our study, we conclude that muskoxen may use an interval timer to schedule their behavioural cycles when forage resources are low, but that the importance and duration of this timer are reduced once environmental conditions allow energetic reserves to be replenished ad libitum. We argue that alimentary function and metabolic requirements are critical determinants of biological rhythmicity in muskoxen, which probably applies to ruminating herbivores in general.

## 1. Introduction

Most organisms exhibit strong temporal organization of behaviour and physiological functions, probably driven by fitness benefits [1,2]. Indeed, coordination of behavioural activities can cycle predictively over timescales (annual, seasonal, daily and hourly) and is typically influenced by exogenous conditions and/or endogenous stimuli. Selective pressures originating from temporal variation in environmental conditions have led to the evolution of internal timing mechanisms (e.g. circadian and circannual rhythms) [3]. Such timing mechanisms allow organisms to predict key environmental changes and synchronize many physiological and behavioural processes essential to fitness, such as cycles of growth, metabolism, fattening and weight loss, hibernation, migration and sexual behaviour [4,5]. Earth's 24 h light–dark cycle is generally considered the strongest exogenous signal (zeitgeber) to which organisms synchronize or schedule many of their activities throughout the day [6].

Polar ecosystems are among the most seasonally extreme environments on the planet, where the intensity of light and photoperiod vary greatly depending on the time of year [7]. Compared to temperate and equatorial zones, polar regions lack the prominent daily oscillation between light and dark for parts of the year, culminating in several weeks or months that the sun remains above (mid-summer) or below the horizon (mid-winter). Because of these unique characteristics, polar regions are an ideal system for studies investigating the existence and persistence of rhythmicity in polar vertebrates under continuous photic conditions (see [8] for a detailed review), which is useful for the identification of circadian organization [9]. While some species have been found to maintain strong, circadian (approx. 24 h) rhythmicity in their locomotor activity (polar bears (*Ursus maritimus*) [10]; thick-billed murres (*Uria lomvia*) [11]) or body temperature profiles (arctic ground squirrels (*Spermophilus parryii*) [12,13]), other species appear to completely lose the temporal organization of internal physiological processes and activity in mid-summer and mid-winter (e.g. Svalbard ptarmigan (*Lagopus mutus hyperboreus*) [14]). Contrasting evidence exists for Svalbard reindeer (*Rangifer tarandus platyrhynchus*) as they have been shown to become arrhythmic in their activity profile under continuous photic conditions [15,16], while other findings suggest that rhythms in activity, heart rate and rumen temperature were merely attenuated and that circadian rhythmicity, albeit weak, persisted throughout the polar year [17].

Besides reindeer, weak or complete lack of circadian organization has also been observed in red deer (*Cervus elaphus* [18]) and horses (*Equus ferus caballus* [19]), which gives rise to the notion that some ungulates may not benefit from a circadian clock that regulates their activity patterns. In fact, the emergent circadian paradigm was recently challenged for wild, large mammals in general [9], arguing that metabolic and alimentary constraints on behaviour supersede the evolutionary advantage of circadian timekeeping for such species [9]. This concept is particularly applicable to ruminating ungulates living under low predation pressure because their activity patterns are primarily restricted by food availability/quality and gut size [20], especially during winter [21]. Under such conditions, ruminating ungulates are more likely to follow ultradian cycles (less than 24 h) of alternate foraging and resting/digesting bouts (i.e. an interval timer), interrupted by occasional transit movements to new foraging sites [22]. Clearly, there remains a need to better understand the interplay between physiological restrictions and local environmental conditions in shaping behavioural organization of wild mammals and of ruminating ungulates specifically.

Our objective was to quantify behaviour-based rhythmicity of a large ruminant, the muskox (*Ovibos moschatus*), considered a species of ecological importance in the Arctic tundra [23] and thus far absent from the chronobiological literature. To do so, we employed a pseudo-experimental approach by using a rare time-series dataset of movements made by adult female muskoxen ranging freely on the high-Arctic tundra of northeast Greenland. Our analytical approach combined hidden Markov models (HMMs), to estimate behavioural states, with wavelet analyses and Lomb–Scargle periodogram (LSP)

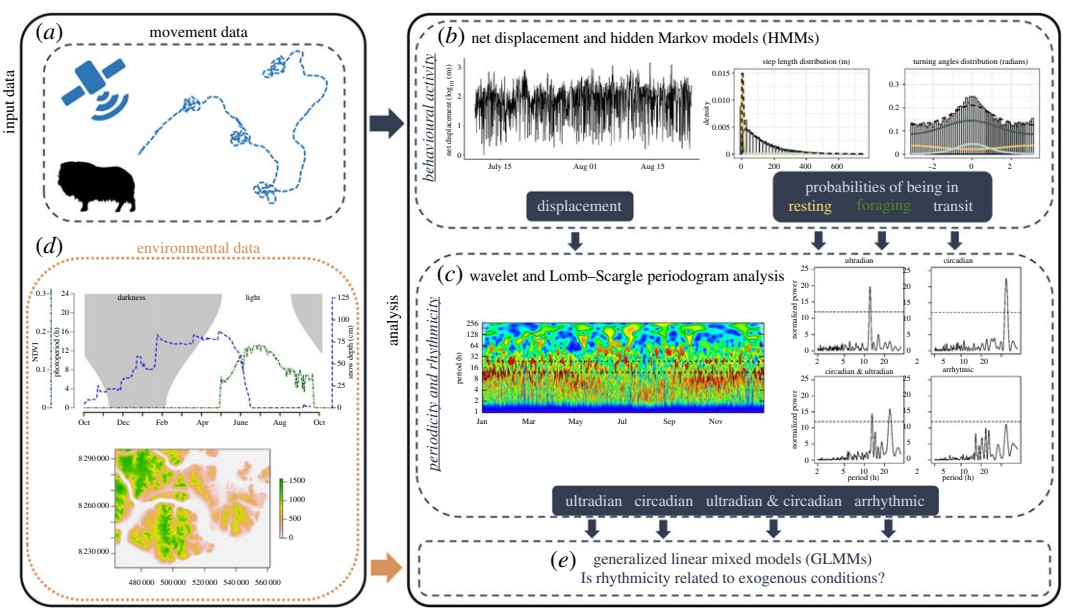

**Figure 1.** Conceptual diagram of the analytical approach used to quantify behavioural-based rhythmicity in muskoxen across the annual cycle of the high-Arctic. Movement data (*a*) were collected by GPS collars scheduled with a 1 h fix rate. Net displacement analysis and hidden Markov models (*b*) were then used to quantify locomotor activity and probabilities of behavioural state occupancy, which served as input data to detect periodicity and classify rhythmicity over time using wavelet and Lomb-Scargle periodogram analysis (*c*). A number of environmental covariates (*d*) were included as fixed effects in generalized mixed effects models (*e*) to assess whether changes in rhythmicity over time were related to exogenous conditions. See the main text for details on the methods employed. Muskox silhouette in (*a*) was taken from http://phylopic.org/.

analysis to quantify chronobiological rhythmicity throughout the polar year (figure 1). Following the recently posited hypothesis for large ruminants [9], we predicted periodicity in behaviour of muskoxen to be driven primarily by foraging and digesting cycles and thus to reflect ultradian rhythmicity (Prediction 1). By contrast, if muskoxen rely strongly on an evolutionary-based endogenous timekeeping mechanism to schedule their biology, as do some other polar species [10,24,25], we would expect circadian rhythmicity in behavioural activity to persist across the annual cycle, including the continuous photic conditions during mid-winter and mid-summer (Prediction 2a). However, behavioural signals can directly track changes in environmental factors, of which the light–dark cycle is the most prominent, possibly masking or reducing the importance of endogenous circadian rhythms [2]. Therefore, we could alternatively expect circadian rhythmicity of muskoxen to occur mainly during periods of the year with a prominent 24 h light–dark cycle with a gradual loss of circadian rhythmicity in behaviour during continuous photic conditions (i.e. mid-winter and mid-summer) as also observed in Svalbard ptarmigan [14] and reindeer [15,16] (Prediction 2b).

# 2. Material and methods

## 2.1. Study area and movement data

The study was initiated in 2013 in northeast Greenland (Zackenberg: 74°28′ N, 20°34′ W) where muskoxen roam across an area that covers approximately 5000 km$^2$ (electronic supplementary material, figure S1A). The annual light cycle at this latitude is characterized by the sun being continuously above the horizon from 30 April to 12 August and continuously below the horizon from 7 November to 2 February. The area holds one of the last remaining indigenous muskox populations in the world, which experiences low predation risk and human disturbance. Muskoxen were captured in Octobers of 2013 and 2015 (see [26,27] for detailed descriptions of capture and handling procedures) and movement data (figure 1*a*) were collected using global positioning system (GPS) collars (Tellus large; Followit Lindesberg AB, Sweden) for a total of 19 adult female muskoxen ($n = 14$ in 2013 and $n = 5$ in 2015). All GPS collars were programmed to record one position per hour and physically impossible movements were removed from the location data prior to analyses [26,28].

A total of 242 378 hourly GPS locations were retained for the statistical analyses (electronic supplementary material, figure S1B), corresponding to 153–1062 days per individual.

## 2.2. Behavioural activity analysis

A commonly used metric in chronobiology to study rhythmicity in animal activity from telemetry data is net displacement [29,30], which we also used here and calculated as the Euclidian distance (m) between hourly locations (log-transformed to stabilize variance). However, net displacement is not a behaviour perse and does not explicitly inform on the range of different behaviours that animals perform, which may need to be considered separately to uncover underlying mechanisms of biological periodicity. Therefore, we also estimated behavioural states from the muskox movement data using HMMs fitted to step lengths (m) and turning angles (radians) between hourly GPS locations (figure 1b). To address potential effects of snow depth on movement variables, separate HMMs were fitted to locations collected during the snow-free and the snow-covered period of the year, as described in detail elsewhere [28]. HMMs were fitted in R v. 3.6.2 [31] via numerical likelihood maximization implemented in the *moveHMM* package [32]. HMMs were constructed to detect two, three, four and five behavioural states without any additional covariates. Inspection of pseudo-residuals and residual autocorrelation revealed that the three-state models were most robust statistically and easily interpreted biologically [28] (see also electronic supplementary material, figures S2 and S3). For each location in the dataset, the HMM-derived probabilities of being in the resting (state 1), foraging (state 2) and transiting (state 3) states were extracted and summarized across weeks and months for all animals to assess the behavioural state budget over time (electronic supplementary material, figure S4). Although we limited state classification to the three most common and straightforward to interpret behavioural states, the HMMs were constructed using an unsupervised approach and not validated by direct observations of the animals. Therefore, state classification should be interpreted with caution and only be considered a proxy for the underlying 'true' behaviour [33]. To maintain a balanced dataset, only individuals with at least 6 or 28 days of GPS data were considered in the weekly- and monthly-scale analyses, respectively.

## 2.3. Periodicity and rhythmicity analyses

Net displacement and the probabilities of behavioural state occupancy as derived from the HMMs were used to detect periodicity over time using wavelet analysis and LSP analysis (figure 1c). Wavelet analyses are particularly useful for identifying circadian and ultradian patterns in behavioural data and for assessing if and how these patterns change or concentrate in a given temporal window [34]. In our case, wavelet analysis was primarily used to search for periodicities in behaviour and displacement considering only those individuals with a complete year of location data ($N = 13$). The main aim was to assess whether periodicities in behaviour and displacement were constant over the year or instead clustered epochs in time, which can be visualized by plotting the wavelet power spectra. Input data for the wavelet analyses were individual-based time series of muskox net displacement and probabilities of behavioural state occupancy at a 1 h resolution using 1 h as the lower Fourier period and 256 h as the upper Fourier period for wavelet decomposition. Wavelet analyses were performed in the R-package *WaveletComp* [35] using the white noise method and 100 bootstrap simulations to test the null hypothesis that there were no periodic structures in the data.

Because location data were unequal across tracked individuals and occasionally contained GPS time-outs (i.e. missed locations; less than 0.1% of the data), the final periodicity tests were performed using the LSP analysis, within the R-package *lomb* [36]. LSP analysis was ideal for our data as it allows for time series with missing data and unequally spaced time series [36]. As for the wavelet analyses, input data for the LSP analyses were individual-based time series of muskox net displacement and probabilities of behavioural state occupancy, but searches for periodicities were performed at the weekly and monthly scale for each individual separately. Although these scales have been used in previous studies [10], we opted for this multi-scale approach to ensure that the results were consistent and unbiased by temporal dependence. For each scale, period lengths between 2 and 18 h (to focus on ultradian rhythmicity) and between 18 and 36 h (to focus on circadian rhythmicity) were considered, following previous studies [10,17]. The output of LSP is the normalized power of periodicity for a given time-series length (i.e. the raw power is divided by twice the total variance in the time series) with a peak in the normalized power indicating a distinct periodicity in activity at the given period (h) [36]. It is possible to detect two distinct peaks in time-series data, which could indicate a true

combination of ultradian and circadian rhythmicity or it could be a data artefact [37]. We followed the protocol of another study to avoid such misclassification of rhythmicity [10]: if two peaks in the normalized power occurred for the same individual within an 18 h window with one peak less than one-third the height of the other, the smaller peak was rejected. Based on the LSP output and using $p < 0.05$ as a threshold for statistically significant peaks in normalized power, each individual-week and individual-month was assigned to one of four rhythmicity classes: ultradian, circadian, ultradian and circadian (two peaks detected), or arrhythmic (no peak detected). Finally, probabilities of transitioning from one rhythmicity class to another were estimated using the Markov chain bootstrap (100 simulations) procedure in the R-package *markovchain* [38].

## 2.4. Statistical analysis

To assess if the probability of being rhythmic (circadian, ultradian, and circadian & ultradian rhythms grouped) was related to exogenous conditions (figure 1*d*), rhythmicity in behaviour was modelled as a binomial (yes or no) variable using the logit link function within generalized linear mixed models (figure 1*e*) with muskox ID nested within year fitted as a random intercept. Separate models were run for rhythmicity in each behavioural state and the net displacement metric. Fixed effects were uncorrelated (Pearson's $|r| < 0.5$) continuous variables including: photoperiod, elevation, proportion of locations in dense vegetation habitat, snow depth and the normalized difference vegetation index (NDVI, a commonly used index for vegetation productivity [39]). Values of fixed effects were calculated for each individual-week or individual-month separately and scaled (mean of zero, one-unit variance) prior to analysis. Photoperiod for the study area was extracted from sunrise and sunset data downloaded from the US Naval Observatory (https://www.usno.navy.mil/USNO/astronomical-applications/data-services), elevation was extracted from the ASTER Global Digital Elevation Model (https://asterweb.jpl.nasa.gov/gdem.asp), and the location of dense vegetation areas was determined using Landsat 4–5TM satellite image [40]. The dynamic variables NDVI and snow depth were extracted from the spatio-temporally explicit SnowModel and MicroMet modelling tools [41,42] and Moderate Resolution Imaging Spectroradiometer (MODIS) daily reflectance data applied to the entire study area and period (see [28,43] for more detail).

# 3. Results

The wavelet analyses revealed statistically significant periodicities in all three behavioural state probabilities and in net displacement, which were most pronounced at periods of 24 h or less; yet periodicities occurred mostly as clustered epochs in time (figure 2). Results of LSP and statistical analyses on the weekly and monthly scales were largely comparable. Therefore, results based on the weekly scale are reported here, while results for the monthly scale are provided in electronic supplementary material, table S1 and figures S5 and S6.

## 3.1. Classification of rhythmicity

The majority of muskox females were classified as rhythmic (LSP peaks of $p < 0.05$, table 1), though patterns varied over time and across behavioural states/activity metric (figure 3). Ultradian rhythmicity (LSP peaks less than 12 h) was most common in the foraging and resting states as well as when analysing net displacement (table 1). Moreover, the probability of remaining in an ultradian rhythm was high (table 2). Circadian rhythmicity (LSP peaks of *ca* 24 h) was most common in the transit state and the probability of remaining in this rhythm was likewise higher than transitioning to another rhythm (tables 1 and 2 and figure 3). LSP analyses also revealed that muskoxen gradually lost rhythmicity in their resting state and in net displacement (peaks of $p > 0.05$), especially between April to September (weeks 16–37), which was also evident yet less pronounced in the foraging state (figure 3). Arrhythmicity in transit behaviour was found occasionally and throughout the year (figure 3). The highest probability to remain arrhythmic in a behavioural state occurred when muskoxen were in the resting state and when analysing net displacement, while the probability of remaining arrhythmic in either the foraging or the transit states was lower (table 2). When in the foraging state, muskoxen were most likely to transition from arrhythmicity to circadian rhythmicity (table 2).

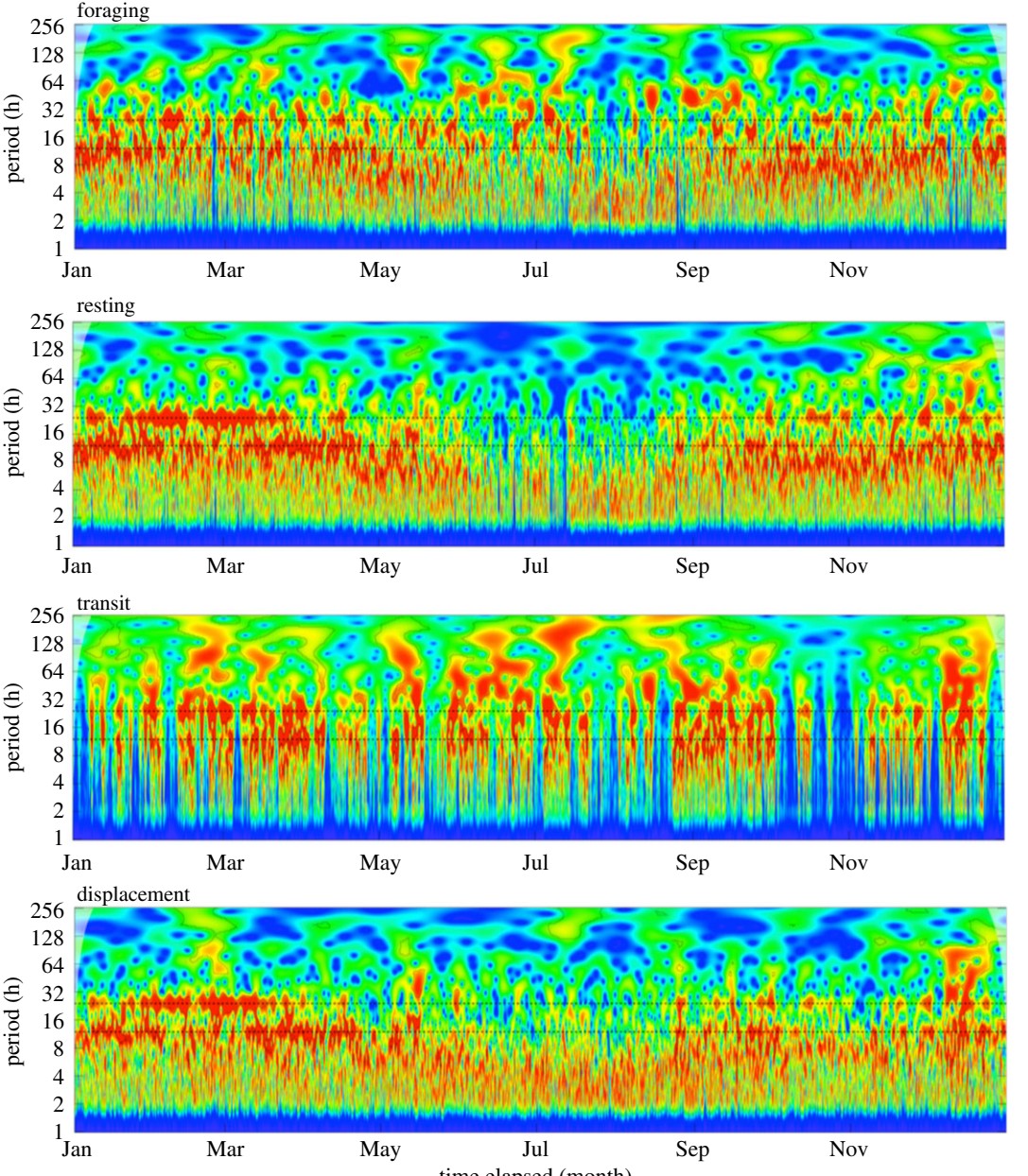

**Figure 2.** Wavelet power spectra of the probability of foraging, resting, transiting and net hourly displacement over an entire year for a single adult female muskox. Times of the year with strong periodicity and of statistical significance at the 5% level are shown in warmer colours (yellow and red). Times of the year without strong periodicity are shown in cooler colours (green and blue). Power values were normalized between panels to facilitate comparison. Horizontal dashed lines are provided at 12 h and 24 h to highlight periodicity at the ultradian and circadian scales, respectively. Wavelet power spectra for all individuals are provided in [44].

## 3.2. Rhythmicity in a changing environment

Rhythmicity in muskoxen was strongly influenced by photoperiod, irrespective of the activity metric analysed (figure 4; electronic supplementary material, table S2). The probabilities of being rhythmic in the foraging and resting states as well as for the net displacement metric were close to 1 during complete darkness (number of daylight hours = 0), but the probabilities decreased significantly as the number of daylight hours increased (figure 4). By contrast, the relation between photoperiod and probability of being rhythmic was positive when muskoxen were in the transit state (figure 4; electronic supplementary material, table S2). A similar divergent relationship was found between snow depth and the probability of being rhythmic, as this was positive for the resting state and the net displacement metric but negative for the transit state (figure 4; electronic supplementary material, table S2). No apparent relationship was found between snow depth and the probability of being

**Table 1.** Overview of classification of rhythmicity for each activity metric on a weekly scale. Absolute and percentage of weeks classified in each rhythm class are provided as well as the corresponding mean ± s.d. peak period (h) as determined using LSP. Note that peak periods for the arrhythmic class were not significant ($p > 0.05$).

| activity metric | rhythmicity class | no. weekly records | % of weekly records | peak period (mean ± s.d. h) |
|---|---|---|---|---|
| foraging | arrhythmic | 174 | 12.6 | 16.9 ± 9.0 |
| | circadian | 338 | 24.4 | 25.9 ± 5.34 |
| | ultradian | 643 | 46.5 | 10.3 ± 2.38 |
| | ultradian and circadian | 228 | 16.5 | 11.6 ± 3.3 & 24.8 ± 4.6 |
| resting | arrhythmic | 293 | 21.2 | 15.2 ± 10.1 |
| | circadian | 89 | 6.5 | 24.4 ± 2.8 |
| | ultradian | 721 | 52.1 | 10.0 ± 2.4 |
| | ultradian and circadian | 280 | 20.2 | 10.5 ± 2.2 & 24.0 ± 1.7 |
| transit | arrhythmic | 209 | 15.1 | 18.9 ± 8.8 |
| | circadian | 635 | 46 | 26.2 ± 5.1 |
| | ultradian | 190 | 13.7 | 12.0 ± 2.1 |
| | ultradian and circadian | 349 | 25.2 | 13.1 ± 2.7 & 24.7 ± 4.7 |
| displacement | arrhythmic | 327 | 23.6 | 15.7 ± 10.3 |
| | circadian | 143 | 10.3 | 24.1 ± 2.3 |
| | ultradian | 699 | 50.6 | 9.9 ± 2.5 |
| | ultradian and circadian | 214 | 15.5 | 10.5 ± 2.3 & 24.0 ± 1.2 |

rhythmic in the foraging state (figure 4; electronic supplementary material, table S2). Instead, a strongly negative relationship was found between NDVI and the probability of being rhythmic in the foraging state, while NDVI had no apparent effect on the probability of rhythmicity for the other behavioural states and the net displacement metric (figure 4; electronic supplementary material, table S2). Following this result, NDVI values over all GPS locations for weeks 23–37 (i.e. end of June to early September, which is roughly the vegetation growing season) were summarized and contrasted between the group of individuals that were classified as rhythmic or arrhythmic while in the foraging state. This revealed that muskoxen with arrhythmic foraging behaviour were in areas with substantially higher mean NDVI values than muskoxen that remained rhythmic in their foraging behaviour during most weeks in the plant growing season (figure 5). Rhythmicity appeared statistically unrelated to elevation and the proportion of locations within dense vegetation habitat (electronic supplementary material, table S2).

## 4. Discussion

Our study clearly shows that behavioural rhythmicity of muskox females changes throughout the polar year and is influenced by the exogenous conditions photoperiod, plant productivity and snow depth. As expected, we found that periodicity in muskox foraging and resting behaviour mostly followed ultradian rhythmicity for large parts of the year (supporting Prediction 1). This finding probably reflects the importance of feeding and digesting cycles in the temporal organization of behaviour by ruminating herbivores [9]. Although we also detected circadian rhythmicity in all behavioural states and net displacement, it did not persist across the annual cycle (figure 3), and was nearly undetected in the resting state during mid-summer (in contrast to Prediction 2a). However, based on the resolution of our data and non-experimental study design, we cannot reject the hypothesis that muskoxen rely on internal/circadian timekeeping mechanisms to schedule their biology as this may have been obscured by strong behavioural responses to environmental factors (i.e. masking effect), a process that complicates inference on the importance and functioning of circadian clocks under natural conditions [45]. Yet, the clear attenuation of circadian rhythmicity in behavioural states and net displacement during the continuous photic conditions of mid-summer provides partial support for

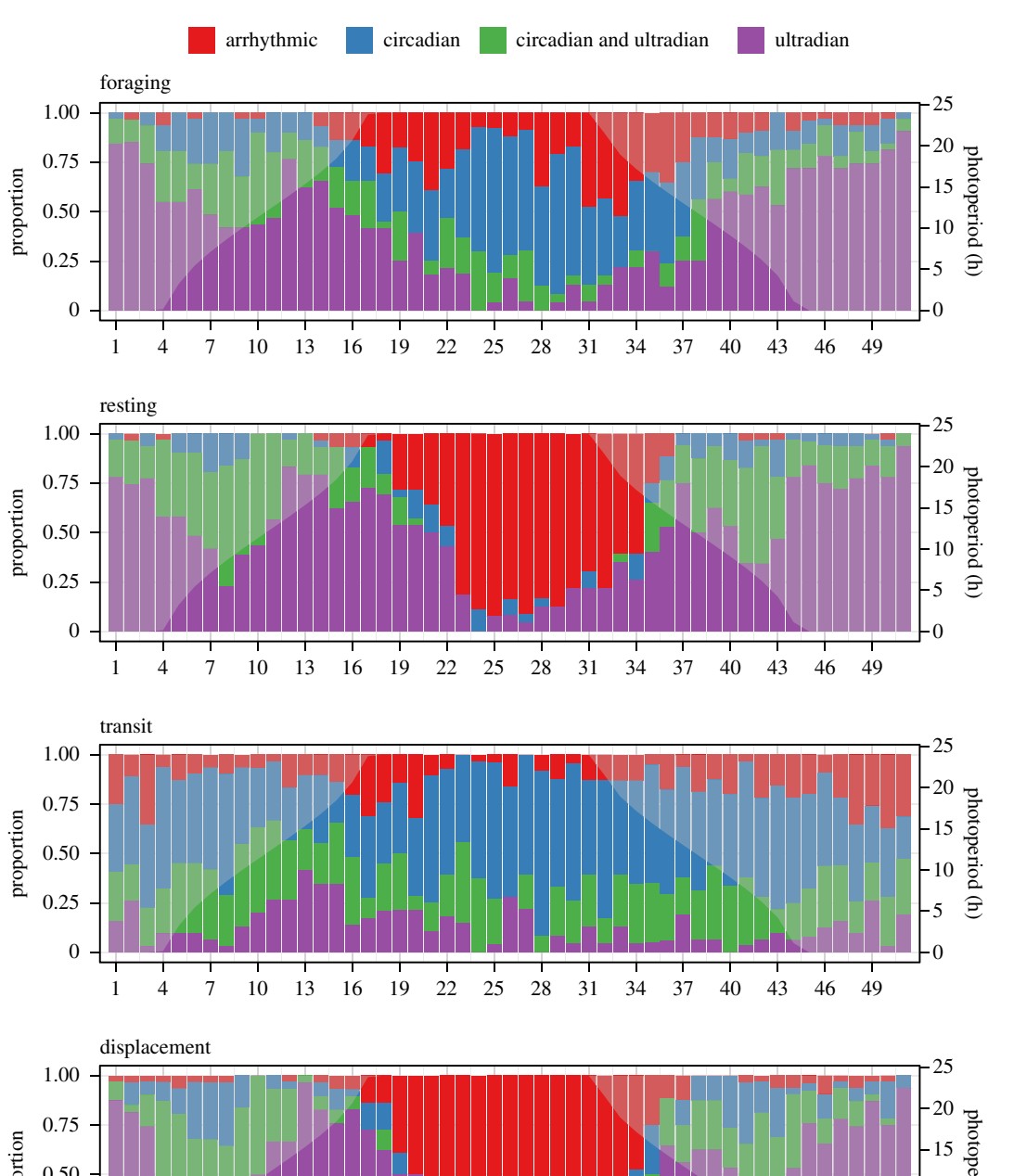

**Figure 3.** Proportion of adult female muskox in each behaviour-based rhythmicity class as determined by Lomb–Scargle periodogram analysis on a weekly time scale, where week 1 is the first week of the year. Rhythmicity classes included: ultradian (peak detected in period 2–18 h only), circadian (peak detected in period 18–36 h only), ultradian & circadian (peaks detected in period 2–18 h and 18–36 h), or arrhythmic (no significant peak detected). Photoperiod (hours of daylight) over the entire year is plotted on top of each panel as light grey.

prediction 2b. Moreover, we found a loss of temporal organization in foraging behaviour, though not resting behaviour, that coincided with increased plant productivity during the snow-free season (figure 4), with individuals classified as arrhythmic in their foraging behaviour generally residing in areas with higher plant productivity than individuals with rhythmic feeding cycles (figure 5). Combined, our results seem to suggest that the scheduling of behaviour by muskoxen may be best explained by an interval timer dictating clear ultradian feeding and rumination cycles when forage

**Table 2.** Transition probabilities in rhythmicity as determined by a discrete Markov chain bootstrap procedure using 1000 simulations for each activity metric separately. Diagonal values indicate the probability to remain within the same rhythm.

| | arrhythmic | circadian and ultradian | circadian | ultradian |
|---|---|---|---|---|
| *foraging* | | | | |
| arrhythmic | 0.18 | 0.13 | 0.35 | 0.33 |
| circadian and ultradian | 0.10 | 0.23 | 0.22 | 0.45 |
| circadian | 0.17 | 0.15 | 0.37 | 0.31 |
| ultradian | 0.09 | 0.16 | 0.16 | 0.59 |
| *resting* | | | | |
| arrhythmic | 0.68 | 0.03 | 0.06 | 0.22 |
| circadian and ultradian | 0.02 | 0.34 | 0.08 | 0.55 |
| circadian | 0.20 | 0.32 | 0.09 | 0.39 |
| ultradian | 0.10 | 0.20 | 0.05 | 0.65 |
| *transit* | | | | |
| arrhythmic | 0.20 | 0.27 | 0.40 | 0.12 |
| circadian and ultradian | 0.16 | 0.21 | 0.46 | 0.17 |
| circadian | 0.13 | 0.24 | 0.50 | 0.13 |
| ultradian | 0.16 | 0.33 | 0.39 | 0.12 |
| *displacement* | | | | |
| arrhythmic | 0.64 | 0.02 | 0.05 | 0.28 |
| circadian and ultradian | 0.06 | 0.30 | 0.12 | 0.52 |
| circadian | 0.13 | 0.24 | 0.19 | 0.43 |
| ultradian | 0.13 | 0.15 | 0.10 | 0.62 |

resources are low and of poor quality, but that the importance and duration of this timer is reduced once environmental conditions allow for build-up of energetic reserves.

Adjustments in rhythmicity and daily timing of activity following changes in food availability are sometimes observed in mammals living at subpolar latitudes [46,47]. Similarly, we found a strong decline in the probability of rhythmicity in muskox foraging behaviour during mid-summer when plant productivity peaked (figure 4). Because muskoxen are capital breeders, they depend heavily on fat stores gained during the short plant growing season to survive and reproduce in the following snow-covered period [48,49]. As such, loss of rhythmicity when food was abundant and constantly available probably emerged from highly irregular feeding and digesting cycles where muskoxen continuously aim to fill their gut with as much high-energy forage as their digestive tract allows, limited in time only by available space in the rumen. By doing so the importance and duration of the interval timer that regulates foraging–resting cycles in winter is reduced, leading to a dampening or even loss of strict ultradian foraging–resting rhythms. However, not all muskox females became arrhythmic in their behaviour in mid-summer despite also occupying sites with relatively abundant and high-quality forage. While the underlying mechanism for this finding remains unclear, a possible ecological explanation could be related to differences in reproductive status between individuals with rhythmic and arrhythmic behaviour. Maintaining rhythmic behaviour could be more important for females that need to care for a calf at heel, while this may be less important for barren females. Unfortunately, we lack data on the reproductive history of most collared muskoxen, which makes it impossible to test this hypothesis here. A possible evolutionary-based explanation as to why some individuals remained behaviourally rhythmic and other did not is the presence of multiple phenotypes with distinct biological rhythms (i.e. chronotypes) within populations. This can alleviate inter- and intra-specific competition and ensure population persistence [2,50], similar to the evolutionary processes responsible for partial migration in some ungulate populations [51]. However, muskoxen are the only large herbivore in our study area and intra-specific competition for food

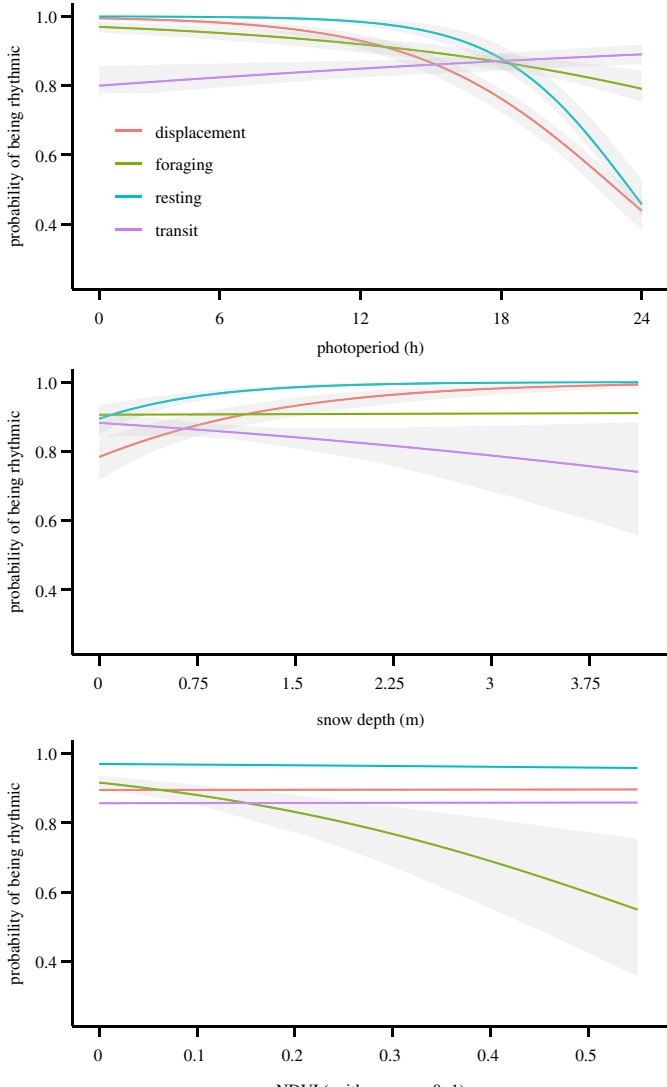

**Figure 4.** Plot of the weekly scale predicted probability of being rhythmic (circadian, ultradian, and circadian & ultradian rhythms grouped) in the foraging, resting and transit state and for net displacement as a function of photoperiod (h of daylight), snow depth (m) and NDVI (unitless; range 0–1). Solid lines show the mean predicted value with shaded grey areas representing the 95% confidence interval, which were generated using a bootstrap procedure with 100 simulations. Non-significant (p > 0.05) relationships are plotted without 95% confidence intervals. Predictions for each relationship were made while keeping other variables in the model (electronic supplementary material, table S2) constant at their mean value.

during the snow-free season is unlikely to be high, given the generally low annual population density [52] and nearly unlimited food base during this time of year [53]. Alternatively, it is possible that individuals with arrhythmic behaviour emerged from the snow-covered period with lower overall body condition and energetic reserves, and were strongly selecting for sites with maximum plant productivity. Indeed, during the long snow-covered season, vegetation is typically access-constrained and of low energetic value [54]. The low quality of forage resources during this time of year could also be a driver for the detected persistence of ultradian rhythmicity while in a foraging state, as few but regular feeding bouts are needed to maintain rumen function [55]. Overall, our study lends support to the recent claim that metabolic requirements and alimentary function are critical determinants of behavioural activity in large wild mammals, limiting the importance of circadian organization in such organisms [9].

Considering behaviour directly in periodicity analyses as done here can facilitate the exploration of critical questions that lie at the interface of ecological and chronobiological research [50]. A much-needed next step of inquiry is to study possible linkages between behaviourally explicit rhythmicity over time and space, and individual fitness. For example, there is evidence from laboratory conditions that

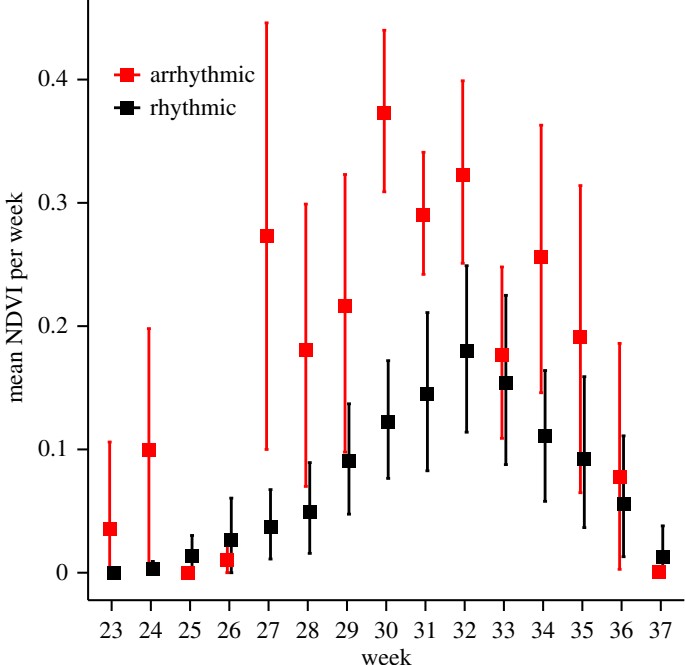

**Figure 5.** Weekly mean and 95% confidence interval of NDVI (unitless; range 0–1) of individuals that were classified as being rhythmic while in the foraging state (circadian, ultradian, and circadian & ultradian rhythms grouped) compared to individuals that were classified as being arrhythmic based on Lomb–Scargle periodogram analyses. The period plotted ranges roughly from end of June to early September.

disrupting rhythmicity has negative effects on individual fitness of parasitic species [56] and that fitness of bacteria decreases when rhythms do not align with the environment [57]. Whether such fitness consequences also exist for other species in natural landscapes remains poorly documented. Based on our results, however, arrhythmicity in foraging behaviour occurs mainly in areas with high plant productivity, which is unlikely to induce a fitness cost. The opposite, i.e. arrhythmicity in areas of low food abundance, might more easily lead to a decrease in health and fitness, but a formal test of this hypothesis is still lacking for large mammals. Fortunately, rapid technological developments in the field of biologging now allow for simultaneous tracking of fine-scale behaviour (e.g. using accelerometer data) and physiological functions (e.g. body temperature) in individual animals [58]. This facilitates not only more thorough quantification of e.g. behavioural states and rumination cycles but also research into rhythm–fitness relationships and future assessments of how the modification of behavioural periodicity due to climate change might influence polar individuals and populations.

Ethics. Approval of sedation and handling of muskoxen were granted under the research permits issued by the Greenland Government, Ministry of Domestic Affairs, Nature and Environment (j. nos. G13-029 and G15-019).

Data accessibility. Data, supporting figures and code are available from the Dryad Digital Repository at https://dx.doi.org/10.5061/dryad.w3r2280n5 [44].

Authors' contributions. F.M.v.B., N.P.H. and N.M.S. designed the study. N.M.S. was responsible for the GPS data collection. L.T.B. ran the HMMs and F.M.v.B. ran the remaining statistical analyses with input from all authors. S.H.P. produced and extracted the environmental data. F.M.v.B. wrote the manuscript with critical input from all authors.

Competing interests. The authors declare no competing interests.

Funding. This study was funded by the Danish charity foundation 15. Juni Fonden, the Danish Environmental Protection Agency and the AUFF Starting grant no. AUFF-F-2o16-FLS-8-16 of F.M.v.B.

Acknowledgements. We thank Aarhus University, Denmark, for logistical support at Zackenberg.

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
