## [Reviewer comments · Royal Society Open Science]

Review History

RSOS-201614.R0 (Original submission)

Review form: Reviewer 1

Is the manuscript scientifically sound in its present form?

Yes

Are the interpretations and conclusions justified by the results?

Yes

Is the language acceptable?

Yes

Do you have any ethical concerns with this paper?

No

Have you any concerns about statistical analyses in this paper?

No

Recommendation?

Accept as is

Comments to the Author(s)

I commend the authors for their thorough revision and I have no further concerns or suggestions.

Review form: Reviewer 2**Is the manuscript scientifically sound in its present form?**

Yes

Are the interpretations and conclusions justified by the results?

Yes

Is the language acceptable?

Yes

Do you have any ethical concerns with this paper?

No

Have you any concerns about statistical analyses in this paper?

No

Recommendation?

Accept with minor revision (please list in comments)

Comments to the Author(s)

I reviewed the previous version of this manuscript for Proc B, and I applaud the authors on producing an excellent revision. I appreciate the attention to detail, and their effort to rigorously address all of my comments from the prior review. I believe the manuscript is scientifically sound and makes an important contribution to the fields of ecology and chronobiology. I have no further major issues to report. Below I have listed a small number of minor grammatical issues that I noticed during my read of the manuscript. I appreciate the opportunity to review this manuscript, and I look forward to seeing it published.

Minor Comments

1. Line 23: I assume the authors meant to refer to the circadian organization of herbivore behavior here, not the circadian organization of herbivores?
2. Line 65: I would suggest using a less colloquial term than 'breeding ground' here. Perhaps something like "...polar regions are an ideal system for studies investigating..."
3. Line 173: Either 'analysis is' or 'analyses are'. Also, change 'to identify' to 'for identifying', and 'to assess' to 'for assessing'.
4. Line 189: I would suggest replacing 'the ideal candidate' with 'ideal'.
5. Line 198: Reword to '...were considered, following previous studies'.
6. Line 281: Replace 'rhythmicity' with 'rhythmic'.
7. Line 375: Suggest removing the (polar) parenthetical here.
8. Line 585: Replace 'was collected' with 'were collected'.

Decision letter (RSOS-201614.R0)

Dear Dr van Beest

On behalf of the Editors, we are pleased to inform you that your Manuscript RSOS-201614 "Environmental conditions alter behavioural organization and rhythmicity of a large Arctic ruminant across the annual cycle" has been accepted for publication in Royal Society Open Science subject to minor revision in accordance with the referees' reports. Please find the referees' comments along with any feedback from the Editors below my signature.

Please submit your revised manuscript and required files (see below) no later than 7 days from today's (ie 06-Oct-2020) date. Note: the ScholarOne system will 'lock' if submission of the revision is attempted 7 or more days after the deadline. If you do not think you will be able to meet this deadline please contact the editorial office immediately.

on behalf of Dr Claudia Wascher (Associate Editor) and Pete Smith (Subject Editor)
openscience@royalsociety.org

Reviewer comments to Author:
Reviewer: 1

Comments to the Author(s)
I commend the authors for their thorough revision and I have no further concerns or suggestions.

Reviewer: 2

Comments to the Author(s)
I reviewed the previous version of this manuscript for Proc B, and I applaud the authors on producing an excellent revision. I appreciate the attention to detail, and their effort to rigorously address all of my comments from the prior review. I believe the manuscript is scientifically sound

and makes an important contribution to the fields of ecology and chronobiology. I have no further major issues to report. Below I have listed a small number of minor grammatical issues that I noticed during my read of the manuscript. I appreciate the opportunity to review this manuscript, and I look forward to seeing it published.

Minor Comments

1. Line 23: I assume the authors meant to refer to the circadian organization of herbivore behavior here, not the circadian organization of herbivores?
2. Line 65: I would suggest using a less colloquial term than 'breeding ground' here. Perhaps something like "...polar regions are an ideal system for studies investigating..."
3. Line 173: Either 'analysis is' or 'analyses are'. Also, change 'to identify' to 'for identifying', and 'to assess' to 'for assessing'.
4. Line 189: I would suggest replacing 'the ideal candidate' with 'ideal'.
5. Line 198: Reword to '...were considered, following previous studies'.
6. Line 281: Replace 'rhythmicity' with 'rhythmic'.
7. Line 375: Suggest removing the (polar) parenthetical here.
8. Line 585: Replace 'was collected' with 'were collected'.

===PREPARING YOUR MANUSCRIPT===

===PREPARING YOUR REVISION IN SCHOLARONE===

Author's Response to Decision Letter for (RSOS-201614.R0)

See Appendix A.

Decision letter (RSOS-201614.R1)

Dear Dr van Beest,

It is a pleasure to accept your manuscript entitled "Environmental conditions alter behavioural organization and rhythmicity of a large Arctic ruminant across the annual cycle" in its current form for publication in Royal Society Open Science.

on behalf of Dr Claudia Wascher (Associate Editor) and Pete Smith (Subject Editor)
openscience@royalsociety.org

Appendix A

Reviewer: 1: Comments to the Author(s)

I commend the authors for their thorough revision and I have no further concerns or suggestions.

***Response: We are pleased that our revised manuscript was well received by the reviewer.*

Reviewer: 2: Comments to the Author(s)

I reviewed the previous version of this manuscript for Proc B, and I applaud the authors on producing an excellent revision. I appreciate the attention to detail, and their effort to rigorously address all of my comments from the prior review. I believe the manuscript is scientifically sound and makes an important contribution to the fields of ecology and chronobiology. I have no further major issues to report. Below I have listed a small number of minor grammatical issues that I noticed during my read of the manuscript. I appreciate the opportunity to review this manuscript, and I look forward to seeing it published.

***Response: We are pleased that our revised manuscript was well received and we thank the reviewer for the suggestions below, which we have all incorporated into the final manuscript draft.*

Minor Comments

1. Line 23: I assume the authors meant to refer to the circadian organization of herbivore behavior here, not the circadian organization of herbivores?

***Response: Correct, we have rephrased the sentence accordingly.*

2. Line 65: I would suggest using a less colloquial term than ‘breeding ground’ here. Perhaps something like “...polar regions are an ideal system for studies investigating...”

***Response: Changed as suggested.*

3. Line 173: Either ‘analysis is’ or ‘analyses are’. Also, change ‘to identify’ to ‘for identifying’, and ‘to assess’ to ‘for assessing’.

***Response: The wording in this sentence has been changed as requested.*

4. Line 189: I would suggest replacing ‘the ideal candidate’ with ‘ideal’.

***Response: Changed as suggested.*

5. Line 198: Reword to ‘...were considered, following previous studies’.

***Response: Changed as requested.*

6. Line 281: Replace ‘rhythmicity’ with ‘rhythmic’.

***Response: Well-spotted, we have changed it accordingly.*

7. Line 375: Suggest removing the (polar) parenthetical here.

***Response: Parenthesis have been removed as suggested*

8. Line 585: Replace ‘was collected’ with ‘were collected’.

***Response: Changed as requested*